# The Expressive Limits of Diagonal SSMs for State-Tracking

**Mehran Shakerinava**[*,×,ℓ] **Behnoush Khavari**[*,×,ℓ]
**Siamak Ravanbakhsh**[×,ℓ] **Sarath Chandar**[×,ℓ,⋉]
Equal contribution[*]  School of Computer Science, McGill University[×]  Chandar Research Lab[×]
Mila – Quebec AI Institute[ℓ]  Polytechnique Montréal[⋉]
mehranshakerinava@gmail.com    khavarib@mila.quebec

## Abstract

State-Space Models (SSMs) have recently been shown to achieve strong empirical performance on a variety of long-range sequence modeling tasks while remaining efficient and highly-parallelizable. However, the theoretical understanding of their expressive power remains limited. In this work, we study the expressivity of input-Dependent Complex-valued Diagonal (DCD) SSMs on sequential state-tracking tasks. We show that single-layer DCD SSMs cannot express state-tracking of any non-Abelian group at finite precision. More generally, we show that $k$-layer DCD SSMs can express state-tracking of a group if and only if that group has a subnormal series of length $k$, with Abelian factors. That is, we identify the precise expressivity range of $k$-layer DCD SSMs within the solvable groups. Empirically, we find that multi-layer models often fail to learn state-tracking for non-Abelian groups, highlighting a gap between expressivity and learnability.

## 1 Introduction

Alternative architectures to Transformers are often motivated by efficiency and computational cost. Equally important, however, is the need to understand their failure modes. Addressing these failures is key to designing better models and requires analyzing three aspects: (1) the model's intrinsic expressive capacity, (2) whether standard learning algorithms (*e.g.,* gradient descent on finite data) can reliably realize solutions within that capacity, and (3) the extent to which these limitations actually transfer to or predict performance on real-world tasks. In this work, we focus on the first aspect, architectural expressivity, and provide empirical observations for the second aspect.

A particularly illustrative class of tasks where Transformers are known to fail is state-tracking (Deletang et al., 2023; Liu et al., 2023; Hahn & Rofin, 2024; Bhattamishra et al., 2022). These tasks are closely related to regular languages in formal language theory and include simple tasks like parity and modular addition. State-tracking tasks are considered representative of a model's performance on real-world problems, such as code execution and program analysis. Examples of Transformers failing to generalize to out-of-distribution (OOD) inputs highlight how limited expressivity can lead models to rely on shortcut solutions that do not generalize beyond the training distribution (Liu et al., 2023). An example of a sequence modeling task that requires keeping track of a state that is being manipulated is program state analysis from big-bench (GoogleResearch, 2021). For example, given the following code,

$$x, y, z = 0, 1, 2; \quad x, y = y, x; \quad y, z = z, y; \quad x, y = y, x$$

what is the value of $x$ after the fourth command? This specific example requires the ability to model permutations of three objects, *i.e.,* the group $S_3$.

State Space Models (SSMs) have emerged as efficient alternatives to Transformers, promising linear-time sequence modeling with recurrent state representations (Gu et al., 2022b). Initially, they were expected to outperform Transformers on state-tracking tasks, because of their recurrent statefulness. However, Merrill et al. (2024) showed that, despite having explicit state representations, SSMs still fail on these tasks, much like Transformers. They highlight this issue for time-invariant or diagonal SSMs in the context of tracking states over sequences of non-solvable group operations. Later

work by Sarrof et al. (2024); Grazzi et al. (2025) reveals that these models are unable to track even solvable groups such as parity and modular counting, due to design limitations in their state transition matrices. We defer a detailed discussion of these findings to Section 3.

In this work, we study the expressivity of input-Dependent Complex-valued Diagonal (DCD) SSMs and show that a single-layer DCD SSM cannot track any non-Abelian group at finite precision. We then show that stacking additional diagonal layers allows the model to track a group if and only if that group has a subnormal series of length equal to the depth of the SSM, with Abelian factors. This result establishes the strict benefit of depth for modeling non-Abelian groups with diagonal SSMs and identifies the precise expressivity range of $k$-layer DCD SSMs within the solvable groups.

Finally, we empirically evaluate diagonal SSMs, both single- and multi-layer, on a range of group state-tracking tasks, including Abelian groups, such as parity ($C_2$) and mod-60 addition ($C_{60}$), as well as non-Abelian groups, such as permutations of 3 elements ($S_3$). In practice, we observe a clear gap between expressivity and learnability of multi-layer models on non-Abelian groups. This suggests that, although some of these models are theoretically expressive enough, they encounter training challenges with standard gradient-based optimization.

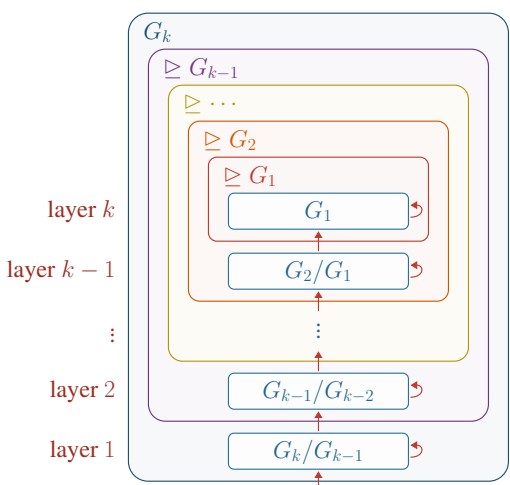

Figure 1: A group $G_k$ can be tracked by a $k$-layer DCD SSM iff the group has a subnormal chain of length $k$ with Abelian factors. Each layer tracks one such Abelian factor as shown in the diagram.

## 2 BACKGROUND

### 2.1 SSMs

We review SSMs and provide background on some of the variants that we refer to in this paper. We begin by defining an SSM layer using notation inspired by previous works (Sarrof et al., 2024; Grazzi et al., 2025).

**Definition 1** (SSM Layer). A $d$-dimensional SSM layer is a parametrized function that takes as input a sequence of $x_t \in \mathbb{F}^n$ and outputs a sequence of $y_t \in \mathbb{F}^m$ via an affine recurrence:

$$h_t = A(x_t)h_{t-1} + b(x_t), \quad (1)$$
$$y_t = \text{dec}(h_t, x_t), \quad (2)$$

where $h_t \in \mathbb{F}^d$ is the state, $A(x_t) \in \mathbb{F}^{d \times d}$ is the transition matrix, $b(x_t) \in \mathbb{F}^d$ is the input vector, and $\text{dec} : \mathbb{F}^d \times \mathbb{F}^n \to \mathbb{F}^m$ is the decoder. The learnable components of the SSM layer are $A$, $b$, dec, and possibly the initial state $h_0$. If $A(x)$ is diagonal for all $x \in \mathbb{F}^n$, we say the SSM layer is diagonal. If $\mathbb{F} = \mathbb{R}$ and $A(x)$ has only real eigenvalues, we say the SSM layer is real. Otherwise, we say the SSM layer is complex.

**Computation** The state's affine recurrence can be efficiently parallelized with the parallel scan (*a.k.a.,* prefix sum) algorithm (Blelloch, 1990) to run in depth $O(\log T)$, as opposed to the naive $O(T)$, where $T$ is the sequence length. The parallel scan algorithm leverages the fact that the composition of affine maps simplifies into another affine map to compute the states in parallel. This is particularly efficient when $A(x)$ is diagonal, as matrix-matrix and matrix-vector multiplications reduce to element-wise multiplications.

Most variants of SSMs are captured by Definition 1. We review some of the most relevant ones.

| Model | $A(x)$ | $B(x)$ | $\text{dec}(h, x)$ |
|---|---|---|---|
| S4 | $\Lambda$ | $Bx$ | $\sigma(\Re(Ch) + Dx)$ |
| S4D | $\Lambda$ (diagonal) | $Bx$ | $\sigma(\Re(Ch) + Dx)$ |
| Mamba | $\exp(\Delta(x) \odot \Lambda)$ | $\Lambda^{-1}\big(\exp(\Delta(x) \odot \Lambda) - I\big)Bx$ | $\sigma(C(x)h + D(x))$ |
| Negative Mamba | $2\exp(\Delta(x) \odot \Lambda) - I$ | $\Lambda^{-1}\big(\exp(\Delta(x) \odot \Lambda) - I\big)Bx$ | $\sigma(C(x)h + D(x))$ |
| AUSSM | $\exp(i\Delta(x) \odot \Lambda(x))$ | $\Delta(x)Bx$ | $\sigma(C(x)h + D(x))$ |

Table 1: Summary of SSM variants in terms of their transition matrix $A(x)$, input vector $B(x)$, and decoder $\text{dec}(h, x)$.

**S4**  The Structured State Space Sequential model (S4) (Gu et al., 2022b) is based on continuous-time linear time-invariant (LTI) state-space models from control theory. It is obtained from Definition 1 by setting $\mathbb{F} = \mathbb{C}$, $A(x) = \Lambda$, $b(x) = Bx$, and $\text{dec}(h, x) = \sigma(\Re(Ch) + Dx)$, where $\Lambda \in \mathbb{C}^{d \times d}$, $B \in \mathbb{C}^{d \times n}$, $C \in \mathbb{C}^{m \times d}$, and $D \in \mathbb{R}^{m \times n}$ are learnable parameters, and $\sigma$ is a nonlinearity.

Careful initialization of the parameters, especially the transition matrix $\Lambda$, *e.g.,* via HiPPO (Gu et al., 2020), allows these models to mitigate the vanishing gradient problem that affects classical RNNs. Moreover, the structure imposed on the $A$ matrix (normal plus low-rank) makes learning efficient. S4 achieved state-of-the-art results on a set of long-range sequence modeling tasks (Tay et al., 2021), where transformers had previously struggled. As a result, it was seen as a promising alternative or complement to attention-based models.

S4 inspired several follow-up models. On the one hand, simpler variants such as DSS (Gupta et al., 2022), S4D (Gu et al., 2022a), and S5 (Smith et al., 2023) simplified S4's architecture while retaining strong performance. On the other hand, more sophisticated models such as H3 (Fu et al., 2023) and Mamba (Gu & Dao, 2024) extended SSMs to handle a more diverse set of tasks, particularly language modeling.

**S4D**  S4D is a simplified version of S4 in which the complex transition matrix $A$ is constrained to be diagonal. This reduces the cost of matrix multiplication in the recurrence equations and hence leads to more efficient computation.

**Mamba**  Designed specifically for language modeling tasks, Mamba introduces input dependence (also called selectivity) in the transition matrix while maintaining the diagonal structure of S4D. It is obtained from Definition 1 by setting $\mathbb{F} = \mathbb{R}$, $A(x) = \exp(\Delta(x) \odot \Lambda)$, $b(x) = \Lambda^{-1}(\exp(\Delta(x) \odot \Lambda) - I)Bx$, and $\text{dec}(h, x) = \sigma(C(x)h + D(x))$, where $\Lambda \in \mathbb{R}^d_{\leq 0}$, $\Delta(x) \in \mathbb{R}^d_{\geq 0}$, $B \in \mathbb{R}^{d \times n}$, $C(x) \in \mathbb{R}^{m \times d}$, and $D(x) \in \mathbb{R}^m$ are learnable functions of the input $x$, and $\sigma$ is a nonlinearity. Note that Mamba's transition matrix $A(x)$ is real-valued and in $(0, 1]$ due to the constraints on $\Lambda$ and $\Delta(x)$. Another variant, due to Grazzi et al. (2025), is negative Mamba which simply replaces $A(x)$ with $2A(x) - I$ to bring the eigenvalue range to $(-1, 1]$.

**AUSSM**  The Adaptive Unitary SSM (AUSSM) (Karuvally et al., 2025) is an input-dependent complex diagonal SSM with unit-modulus transitions. It is obtained from Definition 1 by setting $\mathbb{F} = \mathbb{C}$, $A(x) = \exp(i\Delta(x) \odot \Lambda(x))$, $b(x) = \Delta(x)B(x)$, and $\text{dec}(h, x) = \sigma(C(x)h + D(x))$, where $\Delta(x) \in \mathbb{R}^d_{\geq 0}$, $\Lambda(x) \in \mathbb{R}^d$, $B(x) \in \mathbb{C}^d$, $C(x) \in \mathbb{C}^{m \times d}$, and $D(x) \in \mathbb{C}^m$ are learnable functions of the input $x$, and $\sigma$ is a nonlinearity.

These models have been summarized in Table 1.

## 2.2 Groups, Automata, and State-Tracking

A *semigroup* is a set $G$ equipped with an associative binary operation $\cdot$, where the set is closed under the binary operation. If the operation has an identity element $e$ such that $e \cdot g = g \cdot e = g$ for all $g \in G$, then $G$ is a *monoid*. If every element $g \in G$ has an inverse $g^{-1}$ such that $g \cdot g^{-1} = g^{-1} \cdot g = e$, then $G$ is a *group*. A group is *Abelian* if its operation is commutative, *i.e.*, $g_1 \cdot g_2 = g_2 \cdot g_1$ for all $g_1, g_2 \in G$. A *subgroup* $H \leq G$ is a subset closed under $\cdot$ and inverses (equivalently, a group under the restricted operation). A subgroup $N \trianglelefteq G$ is *normal* if $gNg^{-1} = N$ for all $g \in G$. Note that, by convention,

juxtaposition such as $gN$ or $gNg^{-1}$ denotes the set obtained by multiplying each element of $N$ by $g$ (and by $g^{-1}$ on the right, respectively). When $N \trianglelefteq G$, the set $G/N := \{gN : g \in G\}$ forms the *quotient group*, with operation $(gN) \cdot (hN) = (g \cdot h)N$. A *subnormal series* is a chain

$$(G = G_k) \trianglerighteq G_{k-1} \trianglerighteq ... \trianglerighteq G_1 \trianglerighteq (G_0 = \{e\}). \tag{3}$$

Its *factors* are the quotients $G_{i+1}/G_i$. A group is *solvable* if it admits a subnormal series whose factors are all Abelian. A group is *simple* if it has no non-trivial normal subgroups. See Rotman (2012) for more details.

---

**Example 1.** The group of all permutations of 3 items is denoted $S_3$ and consists of the elements $\{e, (12), (23), (13), (123), (132)\}$. It admits the subnormal series

$$\{e\} \triangleleft C_3 \triangleleft S_3, \tag{4}$$

where $C_3 = \{e, (123), (123)^2\}$ is the cyclic group of order 3. The factors $C_3/\{e\} \cong C_3$ and $S_3/C_3 \cong C_2$ are both Abelian, so $S_3$ is solvable.

---

The set of state-transition functions of a finite automaton when receiving a (possibly empty) sequence of inputs, equipped with composition, forms a monoid. If the automaton is reversible, this monoid is in fact a group. The Krohn–Rhodes theorem (Krohn & Rhodes, 1965) states that any finite automaton can be decomposed into a cascade product of simple groups and reset automata (*i.e.,* flip-flops). An automaton is called solvable if all the simple groups in its decomposition are solvable. In a cascade product of two automata, the state and input of the first automaton provide the input to the second, loosely resembling stacked neural network layers with skip-connections. The Krohn–Rhodes theorem thus reduces the simulation of complex automata to the simulation of simple groups, flip-flops, and their interactions. Since the latter two are relatively easy, the essential challenge lies in the system's ability to simulate groups, which is the focus of this work.

Sequential state-tracking tasks are meant to capture the ability to simulate semigroups. Given a semigroup $G$ its state-tracking task is, given a sequence of elements $x_1, x_2, ..., x_T \in G$, to output a sequence $y_1, y_2, ..., y_T \in G$ such that $y_t = x_1 \cdot x_2 \cdot ... \cdot x_t$. Some specific tasks of interest are parity, which corresponds to the cyclic group $C_2$, and mod-$n$ counting, which corresponds to the cyclic group $C_n$.

## 3 RELATED WORK

Merrill et al. (2024) study the state-tracking capability of SSMs through the lens of circuit complexity. They show that both input-independent non-diagonal and input-dependent diagonal SSMs with real-valued transition matrices belong to the complexity class $\mathsf{TC}^0$, the class of problems solvable by constant-depth, polynomial-size circuits of AND, OR, and threshold gates with unbounded fan-in. This class is widely conjectured to be *incapable of expressing non-solvable state-tracking tasks* such as tracking $S_5$. To address this, they propose either adding nonlinearities to the recurrence, which renders the model non-parallelizable, or making the recurrence non-diagonal and input-dependent. They thus propose an Input-Dependent S4 (IDS4) and empirically show that it performs better on non-solvable state-tracking tasks of relatively longer length. However, the drawbacks of the circuit-complexity approach are that it does not tell us which problems within $\mathsf{TC}^0$ *can* be solved by these SSMs, and it relies on a conjecture in circuit complexity. In this work, we precisely describe the groups that can be tracked with a $k$-layer diagonal SSM (unconditional to any conjecture). In the limit of $k \to \infty$ all solvable groups can be tracked.

While Merrill et al. (2024) focus on the limitations of SSMs on non-solvable groups, Sarrof et al. (2024) highlight a significant limitation on parity, which is the simplest non-trivial solvable group. They prove that input-dependent non-negative diagonal SSMs (*e.g.,* Mamba) cannot solve parity in finite precision for arbitrary sequence lengths. They also show that time-invariant complex-valued diagonal SSMs (*e.g.,* S4D) fail on parity. Connecting with the result of Merrill et al. (2024), this means that common SSM models only cover a very small subset of $\mathsf{TC}^0$. On the positive side, Sarrof et al. (2024) show that a Mamba layer can simulate a flip-flop, and as a result, multi-layer Mamba can simulate counter-free automata.

Extending these results to non-diagonal models, Grazzi et al. (2025) prove that a multilayer SSM can solve parity in finite precision for arbitrary sequence lengths only if at least one layer has a negative eigenvalue. This implies that even DeltaNet[1] fails to solve parity in its standard form. They argue that existing SSMs typically lack either input dependence or negative (resp. complex) eigenvalues. Having both of these properties in some layer is essential for solving parity (resp. modular counting) (Khavari et al., 2025). To address this, they modify Mamba and DeltaNet to allow eigenvalues in the range $[-1, 1]$ instead of $[0, 1]$. This leads to empirical improvements on both parity and real-world tasks.

While negative eigenvalues are sufficient for solving parity, Grazzi et al. (2025) show that complex eigenvalues are necessary for harder tasks such as modular counting. Thus, although modifying Mamba to include negative eigenvalues enables it to solve parity, solving more complex tasks demands transition matrices with complex eigenvalues. They note that such matrices can be constructed by multiplying several real-eigenvalued matrices, provided the product is non-triangular.[2]

Building on this idea, Siems et al. (2025) proposes DeltaProduct, an adaptive extension of DeltaNet that generalizes the transition matrix from diagonal-plus-rank-1 to a structured rank-$n$ matrix. Here $n$ is tunable to trade off between expressivity and efficiency. Their construction is based on products of $n$ generalized Householder matrices. A limitation of this approach is the computational cost of multiplying non-diagonal matrices.

Karuvally et al. (2025) propose the Adaptive Unitary SSM (AUSSM), which is a complex-valued input-dependent diagonal SSM with unit-modulus eigenvalues. They show that a single-layer AUSSM can simulate any Abelian group but it cannot simulate flip-flops. Since Mamba is able to simulate flip-flops, they propose interleaving Mamba and AUSSM layers to handle solvable automata (according to Krohn–Rhodes theory). However, it is currently unknown (1) if a single-layer AUSSM can simulate non-Abelian groups, (2) what the expressive capacity of $k$-layer AUSSM is, and (3) if it is unconditionally true[3] that multi-layer AUSSMs cannot simulate non-solvable groups. These are questions that we aim to address.

## 4 THEORETICAL RESULTS

### 4.1 SINGLE-LAYER DCD SSM

We begin by analyzing the limitations of a single-layer input-Dependent Complex-valued Diagonal (DCD) SSM (recall Definition 1) on sequential state-tracking tasks for groups. We assume that $A$, $b$, and dec are universal function approximators. In this section, we build up to the following theorem.

> **Theorem 1.** *There is a single-layer DCD SSM that tracks $G$ at finite precision iff $G$ is Abelian.*

Having $b$ in the state recurrence makes the problem non-trivial. Without $b$, the state recurrence is given by $h_t = A(x_t)h_{t-1}$ and since $A(x_t)$ is assumed to be diagonal and since diagonal matrices commute, the expressed group operation has to be commutative, *i.e.,* the group is Abelian. However, with $b$, the SSM can perform non-commutative state updates. We show that even with $b$, a single-layer DCD SSM cannot express non-Abelian groups at finite precision. In other words, our result implies that having $b$ does not increase expressivity for tracking groups at finite precision.

Another point that makes the problem difficult is the existence of a decoder (especially a powerful one), as it can implement complex decision boundaries in the state space. For example, we cannot assume that feeding the sequence $(g, g^{-1})$ to the SSM layer results in the identity map on the state. As long as the initial state $h_0$ and the final state $h_2$ decode to the same group element, *i.e.,* $\text{dec}(h_0) = \text{dec}(h_2)$, the SSM is treating the input sequence correctly.

---

[1]DeltaNet is a linear attention model that can also be interpreted as an SSM, with a diagonal plus low-rank transition matrix. More specifically, a generalized Householder matrix.

[2]Because the eigenvalues of a triangular matrix are the diagonal elements, and the product of two triangular matrices remains triangular.

[3]Unconditionally true means that the argument does not depend on any conjecture.

An important property of SSMs (and linear RNNs in general) is that the state update corresponding to a finite *sequence* of inputs simplifies into an affine map. More specifically, given an input sequence $\bar{x} = (x_1, x_2, ..., x_T)$, the state update from $t = 0$ to $t = T$ of a diagonal SSM is given by

$$h_T = \lambda(\bar{x}) \odot h_0 + b(\bar{x}), \tag{5}$$

where

$$\lambda(\bar{x}) := \prod_{i=1}^{T} \lambda(x_i), \tag{6}$$

$$b(\bar{x}) := \sum_{i=1}^{T} \left( \prod_{j=i+1}^{T} \lambda(x_j) \right) b(x_i). \tag{7}$$

This is a key property that we use in our proofs. Importantly, note that we have now overloaded $\lambda$ and $b$ to also accept a *sequence* of inputs.

First, we show that if some input $x$ causes some coordinate $j$ of the state to contract ($|\lambda(x)_j| < 1$), expand ($|\lambda(x)_j| > 1$), or drift ($|\lambda(x)_j| = 1 \wedge b(x) \neq 0$), then we can construct another SSM that still solves the task while keeping state-coordinate $j$ fixed. As a result, such state-coordinates are useless for tracking groups and can be effectively ignored by the decoder.

**Notation.** We sometimes use multiplicative notation for groups. For example, $g^3$ means $g \cdot g \cdot g$ and $gh$ means $g \cdot h$. Sometimes we may want to concatenate group elements to form a sequence. We use parentheses to denote sequences and write $\langle g \rangle$ for a sequence of length one consisting only of $g$. We use multiplicative notation for the concatenation of sequences as well. For example, $\langle g \rangle \langle h \rangle^5$ is a sequence of length six consisting of a single $g$ followed by five $h$ elements.

> **Lemma 1.** *Let $M$ be a single-layer DCD SSM that tracks group $G$ at finite precision with $|\lambda(x)_j| \neq 1$ or $|\lambda(x)_j| = 1 \wedge b(x)_j \neq 0$ for some $x \in G$ and $j \in [d]$, then there exists another single-layer DCD SSM $\widetilde{M}$ that also tracks group $G$ at finite precision with $\tilde{\lambda}(g)_j = 0$ and $\tilde{b}(g)_j = c$ (for some constant $c$) for all $g \in G$.*

A proof is provided in Appendix C.1.

Repeating the lemma above for all coordinates $j \in [d]$ lets us construct another SSM that tracks the same group but has no contracting, expanding, or drifting coordinates. Therefore, we can now assume that all coordinates of the state have neutral rotation dynamics (case 2 in Appendix A) for all inputs. Next, we show that if two inputs have neutral rotation dynamics with *distinct* centers of rotation, then the SSM can diverge. The following lemma is useful.

> **Lemma 2.** *The composition of neutral rotations $h \mapsto \lambda(h - c_1) + c_1$ and $h \mapsto \lambda^*(h - c_2) + c_2$, where $\lambda^*$ is the conjugate of $\lambda$, with distinct centers of rotation $c_1 \neq c_2$ is a non-zero translation.*

A proof is provided in Appendix C.2.

The above lemma gives us a strategy to make certain SSM configurations diverge.

> **Lemma 3.** *If a single-layer DCD SSM tracks group $G$ at finite precision and there exist two inputs $g, h \in G$ that induce neutral rotation about distinct centers in some coordinate $j \in [d]$ of the state, then the SSM diverges on some input sequence.*

A proof is provided in Appendix C.3.

By putting the lemmas together, we can prove Theorem 1. A proof is provided in Appendix C.4.

## 4.2 MULTI-LAYER DCD SSM

In this section, we extend our results to the multi-layer setting, where the input to the $r$th layer is the input token $x$ and the states of the previous $r - 1$ layers, denoted $h^{(1)}, ..., h^{(r-1)}$. We aim to prove the following theorem.

**Theorem 2.** *There is a $k$-layer DCD SSM that tracks $G$ at finite precision iff one can write*

$$(G = G_k) \trianglerighteq G_{k-1} \trianglerighteq ... \trianglerighteq G_1 \trianglerighteq (G_0 = \{e\}) \tag{8}$$

*where $G_{i+1}/G_i$ is Abelian for all $i \in [k]$.*

We begin by applying the same approach as the single-layer case to construct another SSM with simpler state space and state transitions that tracks the same group.

**Lemma 4.** *If a $k$-layer DCD SSM tracks group $G$ at finite precision, then there exists another $k$-layer DCD SSM that also tracks group $G$ at finite precision, where, for all layers $r \in [k]$ and all state-coordinates $j \in [d]$, the transition dynamics is fixed or a neutral rotation about a center that is a function of the states of previous layers ($h^{(1)}, ..., h^{(r-1)}$).*

A proof is provided in Appendix C.5. We then use it to prove the main theorem in Appendix C.6.

## 5 A DIAGONAL SSM FOR $S_3$

In this section, we present an example of a two-layer diagonal SSM that can track the non-commutative yet solvable group $S_3$ for arbitrary sequence lengths. This section has two purposes. First, it provides a concrete illustration of the theory introduced in the previous section. We begin by showing how two relatively simple automata can be combined to form a non-Abelian solvable automaton, and then demonstrate how this construction can be encoded in stacked diagonal SSM layers. Secondly, it highlights the gap between expressivity and learnability. While the theory suggests that a two-layer diagonal SSM can track the non-Abelian group $S_3$, our experiments in the next section show that, in practice, the model often struggles to learn solutions that generalize to longer sequences. This implies that while generalizable solutions would lie within the expressive power of the diagonal SSMs we study, they may be difficult for the learning algorithm to find.

### 5.1 THE $S_3$ GROUP

The symmetric group $S_3$ is the group of all permutations of three elements. It has six elements in total and provides the smallest non-Abelian example of a finite group. The elements can be written in cycle notation as $S_3 = \{e, (12), (13), (23), (123), (132)\}$, where $e$ denotes the identity permutation, the transpositions $(12), (13), (23)$ swap two elements, and the 3-cycles $(123)$ and $(132)$ permute all three elements cyclically in opposite directions. $S_3$ can be decomposed as the semi-direct product of two Abelian groups, $C_2$ and $C_3$. It can also be presented by two generators, for example $S_3 = \langle (12), (123) \rangle$, of orders 2 and 3, *i.e.*, $(12)^2 = e$, $(123)^3 = e$. The Cayley table of group multiplications for $S_3$ is given in Appendix B.1.

### 5.2 DECOMPOSING THE $S_3$ AUTOMATON

Now, we show that there exists a finite state automaton, built as a cascade product of two simpler automata, that can track $S_3$. Figure 2 illustrates one such composition of automata capable of tracking $S_3$. To illustrate how this solution works, we encode each $g \in S_3$ as $s^\alpha r^\beta$, with $s = (12)$, $r = (123)$ denoting swap and rotation generators, and multiplication applied left to right. Under this encoding, $S_3$ can be rewritten as $S_3 = \{e, s, sr^2, sr, r, r^2\}$. The combined automaton operates as follows: $s^\alpha r^\beta$ first applies $\alpha$ swaps on the two-state automaton with state $Q^{(1)} \in \{-1, 1\}$, yielding $Q^{(1)} = Q_0^{(1)}(-1)^\alpha$, with $Q_0^{(1)}$ being the initial state. The second automaton then takes as input both the input group element and the state of the first automaton and transitions its state $Q^{(2)} \in \{Q_1, Q_2, Q_3\}$ according to the transition rule illustrated in Figure 2; that is, if the state of the first automaton is $Q^{(1)} = 1$, upon seeing a group element with $\beta = 1$, it rotates its states according to the cyclic permutation $(123)$, and if $\beta = 2$, the cyclic permutation $(132)$ is applied; however, if $Q^{(1)} = -1$, the cyclic permutations are applied in the other direction, *i.e.*, for $\beta = 1$, the state transitions according to $(132)$ and for $\beta = 2$, it transitions according to $(123)$. In Appendix B.2, we give concrete examples showing how this encoding, together with the transition rule of the two automata, correctly reproduces group multiplication and thus tracks $S_3$ with the compounded

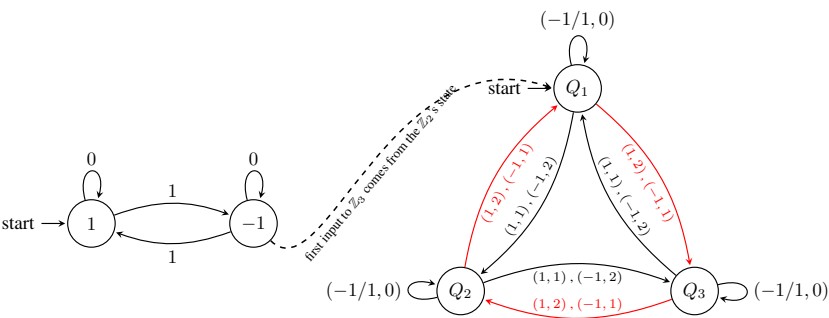

Figure 2: Compound automaton combining (left) two-state parity and (right) three-state cyclic group. The dashed curved line represents the connection between the two cyclic automata, equivalent to the semi-direct product of their groups. It shows that the automaton on the right considers the state of the first automaton, besides the original input. Commas separate inputs that produce the same state transition. States labelled *start* are the initial states of the automata.

automaton. In what follows, we use the result of Example 2 in Appendix B.2 based on this specific encoding, summarized in Table 4, to parameterize a two-layer AUSSM that tracks $S_3$ consistently.

### 5.3 SIMULATING THE 2-AUTOMATON FOR $S_3$ WITH DIAGONAL SSMS

From Theorem 1, each diagonal SSM layer can model one Abelian automaton; it remains to show that we can stack them in a way that simulates the cascade product of the two Abelian automata.

With the encoding $(\alpha, \beta)$ for the input $s^\alpha r^\beta$, for the first AUSSM, with the state equation $h_t = e^{-i\Delta(x)\Lambda(x)}h_{t-1} + B(x_t)$, we set $B(x) = 0$, $\Delta(x) = 1$, and $\Lambda((0, \beta)) = 0$, $\Lambda((1, \beta)) = \pi$.
The second layer gets the input $(h^{(1)}, \alpha, \beta)$, with $h^{(1)}$ being the state of the first AUSSM layer, fed to the second layer with a skip connection. Here we set $B^{(2)}(x) = 0$, $\Delta^{(2)}(x) = 1$, and $\Lambda^{(2)}((1, \alpha, \beta)) = \frac{2\pi}{3}\beta$, $\Lambda^{(2)}((-1, \alpha, \beta)) = -\frac{2\pi}{3}\beta$. The full state of the SSM will be $(h_0^{(1)}e^{-i\alpha\pi}, h_0^{(2)}e^{-\frac{2\pi i}{3}\beta}h_0^{(1)}e^{-i\alpha\pi})$. This results in a finite number of states, and with a correct decoding, we can map the states to the elements of the group.

## 6 EXPERIMENTS

We evaluate single- and two-layer diagonal SSMs on a set of solvable group state-tracking tasks. Our goal is to measure what these models learn with standard training, not just what they can represent in theory. Our models include Mamba (Gu & Dao, 2024), negative Mamba (Grazzi et al., 2025), AUSSM (Karuvally et al., 2025), RNN, and simplified AUSSM where $\Delta$ and $B$ have been removed since they are theoretically not needed for expressing groups.

**Datasets.** Our set of tasks consists of different kinds of solvable groups that we are interested in: parity, which is $C_2$, a small, a medium and a large cyclic Abelian group, $C_6$, $C_{24}$ and $C_{60}$ respectively, two products of cyclic groups, $C_2 \times C_4$ and $C_3 \times C_6$, and two small solvable non-Abelian groups with subnormal chains of length 2, $S_3$ and $A_4$. We also include the non-solvable group $A_5$, which we expect no model can learn.

For a cyclic group $C_N$ with $N \in \mathbb{N}$, the task is essentially addition mod $N$, with the input tokens being chosen uniformly at random from the group. An example input for the task $C_{60}$ is the sequence $[51, 20, 4, 49]$ and the correct output is $[51, 11, 15, 4]$.

The group $S_3$ was introduced in detail in Example 1. The alternating group $A_4$ consists of the 12 even permutations of four elements. Like $S_3$, it admits a subnormal chain of length two with Abelian factors, and is therefore solvable.

The alternating group $A_5$ consists of the 60 even permutations of five items. In contrast to $S_3$ and $A_4$, its shortest subnormal series is $\{e\} \lhd A_5$, with the non-Abelian factor $A_5/\{e\} \cong A_5$. Consequently, $A_5$ is not solvable.

**Experimental Details.** Models were trained on sequences of up to length 60 and tested on sequences up to length 1000 to assess their length generalization performance. We applied curriculum learning by gradually increasing the length of training sequences, starting from length 2. We report the longest sequence length $\geq 100$ at which the trained model can obtain above 90% accuracy. If the model was unable to extrapolate we report ✗. The results can be seen in Table 2.

All experiments were conducted with standard FP32 precision. We performed a grid search over three state dimensions $\{32, 64, 128\}$, three learning rates $\{1e^{-4}, 5e^{-4}, 1e^{-3}\}$, and three learning-rate schedulers $\{\text{fixed}, \text{reduce on plateau}, \text{cosine}\}$. The AdamW optimizer was used with these learning rates and a weight decay of 0.01. Each experiment was run with three random seeds, and we reported the best result across seeds. The embedding and model dimensions were fixed at $m = n = 698$ across all experiments. The choice of a relatively large embedding dimension was motivated by stabilizing optimization across both real- and complex-valued kernels. Unless otherwise noted, models were trained with a batch size of 256. We used gelu nonlinearities between SSM layers and residual connections within each SSM layer. The initial state $h_0$ of the SSMs was set to $\mathbf{1}$ and normalized to unit norm.

| Task | Mamba | Negative Mamba | Simple AUSSM | AUSSM | RNN |
|---|---|---|---|---|---|
| $C_2$ | ✗ | 1000 | 160 | 1000 | 1000 |
| $C_6$ | ✗ | ✗ | 240 | 940 | 1000 |
| $C_{24}$ | ✗ | ✗ | 240 | 260 | 1000 |
| $C_{60}$ | ✗ | ✗ | 300 | 240 | ✗ |
| $C_2 \times C_4$ | ✗ | ✗ | 140 | 200 | 1000 |
| $C_3 \times C_6$ | ✗ | ✗ | 500 | 200 | ✗ |
| $S_3$ | ✗ | ✗ | ✗ | ✗ | 1000 |
| $A_4$ | ✗ | ✗ | ✗ | ✗ | 1000 |
| $A_5$ | ✗ | ✗ | ✗ | ✗ | ✗ |

(a) Single-layer models.

| Task | Mamba | Negative Mamba | Simple AUSSM | AUSSM | RNN |
|---|---|---|---|---|---|
| $C_2$ | ✗ | 1000 | 1000 | 200 | 1000 |
| $C_6$ | ✗ | ✗ | 240 | 100 | 1000 |
| $C_{24}$ | ✗ | ✗ | 300 | 160 | 1000 |
| $C_{60}$ | ✗ | ✗ | 260 | ✗ | ✗ |
| $C_2 \times C_4$ | ✗ | 360 | 160 | ✗ | 1000 |
| $C_3 \times C_6$ | ✗ | ✗ | 260 | 200 | ✗ |
| $S_3$ | ✗ | ✗ | ✗ | ✗ | 1000 |
| $A_4$ | ✗ | ✗ | ✗ | ✗ | 1000 |
| $A_5$ | ✗ | ✗ | ✗ | ✗ | ✗ |

(b) Two-layer models.

Table 2: Performance of various models on state-tracking tasks. Each table reports the longest sequence length $\geq 100$ where a model is able to achieve accuracy greater than 90%. The maximum training length was 60 for all models over all state-tracking tasks. ✗ indicates that the model failed to extrapolate to long sequences.

**Results.** For RNNs, there is no theoretical expressivity barrier that would stop them from successfully learning all tasks. However, in our experiments, the single-layer RNN struggles on $C_{60}$, likely due to the large size of the group, and the two-layer variant struggles on both $C_{60}$ and $A_4$. It is possible that with different architectural hyperparameters or longer training the RNN would have succeeded.

For Mamba, since it does not have negative eigenvalues, in accordance with previous theoretical results (Sarrof et al., 2024), we expect it to not be able to track any group. This is indeed observed in our experiments. Negative Mamba is a modified version of Mamba which allows for negative eigenvalues and with a single layer, is expected to only be able to solve parity (Grazzi et al., 2025). This is also observed in our experiments. On the other hand, according to our theory, stacking two layers, each of which is capable of solving $C_2$ (*e.g.,* Negative Mamba), can do state-tracking for $C_4$[4]. This is because the group $C_4$ can be written as the subnormal chain $C_4 \rhd C_2 \rhd \{e\}$, where the factors are all $C_2$, *i.e.,* Abelian. We see in practice that 2-layer Negative Mamba is able to do state-tracking for $C_2 \times C_4$. Interestingly, this is a case where the increased expressivity from stacking layers is usable through gradient-based learning.

For AUSSM and simple AUSSM, their single-layer variants are theoretically able to express Abelian groups, which in our case are the six groups $C_2$, $C_6$, $C_{24}$, $C_{60}$, $C_2 \times C_4$, and $C_3 \times C_6$. Gradient-based optimization has successfully been able to find solutions that extrapolate in this case as well (with the exception of two anomalies for two-layer AUSSM). However, their two-layer variants have not been able to match the expressive capacity that is expected from them as they fail on $S_3$ and $A_4$.

---

[4]This has also been stated in Theorem 2 of (Grazzi et al., 2025).

Overall, these experiments point to a learnability gap. The main bottleneck of non-Abelian tracking is not expressivity but optimization and the model's built-in bias: the solutions exist in weight space, yet standard training does not reliably reach them.

To gain insight into the potential causes of this learnability issue, we initialized AUSSM near the analytical solution for $S_3$ from Section 5.3. We observed that sufficiently close initialization improves training: the model successfully learns and extrapolates to sequences at least four times longer than those seen during training. This suggests that solutions lie within a basin of attraction in the loss landscape. However, the broader structure of the loss landscape is still not well understood. A study analogous to Hahn & Rofin (2024), which investigates transformers, would be a valuable future direction for SSMs. Notably, since initialization near the solution aids optimization in SSMs, the challenges they face may differ substantially from those of transformers, where solutions have been shown by Hahn & Rofin (2024) to correspond to isolated points in weight space.

## 7 DISCUSSION

Our results reveal a sharp distinction between what diagonal SSMs can represent in principle and what they can learn in practice. Theoretically, a single diagonal layer suffices for all Abelian groups, and stacking layers expands expressivity exactly to solvable groups with a subnormal series of matching length. This characterization places diagonal SSMs within a precise group-theoretic boundary: they are strictly weaker than non-diagonal recurrent models, yet depth provides a disciplined pathway to handle increasingly complex dynamics.

Empirically, however, our experiments show that diagonal SSMs often fail to realize their expressive potential. Even two-layer models, which can provably represent $S_3$, rarely discover solutions that generalize beyond training lengths. This gap points to difficulties of gradient-based optimization when searching for encodings of non-Abelian structure within the restricted hypothesis class. It also suggests that diagonality, while efficient, imposes inductive biases that may actively hinder training on harder tasks.

From a broader perspective, our findings connect to other architectural choices in sequence models. Allowing block-diagonal structure, even in $2 \times 2$ form, would in principle lift the expressivity of SSMs into $NC^1$, enabling simulation of non-solvable groups. Similarly, introducing complex-valued attention weights in Transformers may bring them closer to the expressivity frontier we identify here for SSMs. Finally, it is worth noting that many practical applications do not demand arbitrary-length state-tracking; being able to stably handle moderately long sequences may be sufficient, though our results clarify what is lost at the limit.

Finally, although we state the main results for groups, the framework extends naturally to semigroups and monoids. Allowing inputs that *reset* a layer (*i.e.,* map a coordinate to a fixed center under finite precision) lets a diagonal SSM simulate reset automata in addition to group components. Consequently, the same depth-based view applies to cascade products comprising Abelian group factors and resets, bringing the analysis in line with the Krohn–Rhodes perspective on solvable automata.

## 8 CONCLUSION

We have given a complete characterization of the expressive power of diagonal SSMs on group state-tracking tasks. A single diagonal layer cannot track non-Abelian groups, while a $k$-layer SSM can track precisely those groups admitting a subnormal series of length $k$ with Abelian factors. This establishes a provable expressivity gap between single- and multi-layer diagonal SSMs.

Our experiments further demonstrate a learnability gap: despite their theoretical capacity, multi-layer diagonal SSMs struggle to learn even small non-Abelian solvable groups such as $S_3$ and $A_4$. Together, these results highlight the importance of separating expressivity from learnability when evaluating new sequence architectures. Future progress will require not only expanding the expressive frontier — through, for example, block-diagonal transitions or hybrid models — but also developing training methods that can reliably reach solutions guaranteed to exist in principle.

## 9 ACKNOWLEDGEMENTS

We thank Ashish Sabharwal and Razvan Pascanu for useful discussion and Yusong Wu, Alex Hernandez-Garcia, Gauthier Gidel, and Ioannis Mitliagkas for feedback on an earlier draft of this paper. This research is in part supported by CIFAR AI Chairs and the NSERC Discovery program. Mila and the Digital Research Alliance of Canada provided computational resources.

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

## A  BACKGROUND: ONE-DIMENSIONAL COMPLEX AFFINE DYNAMICS

We consider the affine recurrence

$$x_{t+1} \;=\; \lambda\, x_t + b, \qquad \lambda, b \in \mathbb{C}, \quad t = 0, 1, 2, \dots \tag{9}$$

with initial condition $x_0 \in \mathbb{C}$. This section records a complete, self-contained classification of the dynamics and then isolates the bounded regimes relevant for our work.

**Closed form and fixed points.**  Iterating (9) yields

$$x_t \;=\; \lambda^t x_0 \;+\; b \sum_{k=0}^{t-1} \lambda^k \;=\; \begin{cases} \lambda^t\big(x_0 - c\big) + c, & \lambda \neq 1, \\ x_0 + t\,b, & \lambda = 1, \end{cases} \qquad c := \frac{b}{1-\lambda} \; (\lambda \neq 1). \tag{10}$$

A fixed point exists iff either $\lambda \neq 1$ (unique fixed point $c$) or ($\lambda = 1$ and $b = 0$) (every point is fixed). When $\lambda = 1$ and $b \neq 0$ there is no fixed point.

**Shift-of-origin reduction.**  When $\lambda \neq 1$, define the shift $y_t := x_t - c$ with $c = \dfrac{b}{1-\lambda}$. Then

$$y_{t+1} = \lambda y_t, \tag{11}$$

so the inhomogeneous recurrence (9) is conjugate to the homogeneous linear map $y \mapsto \lambda y$. All asymptotics therefore reduce to the magnitude and argument of $\lambda$:

**Complete case split.**  Let $\lambda = r e^{i\theta}$ with $r = |\lambda|$.

1. *Strict contraction* ($|\lambda| < 1$): $x_t \to c$ exponentially at rate $r^t$ (independent of $x_0$). If $\lambda \in \mathbb{R}_{>0}$ there is no rotation; otherwise each step rotates by $\theta$ while contracting.

2. *Neutral rotation* ($|\lambda| = 1, \lambda \neq 1$): $|x_t - c| = |x_0 - c|$; the orbit is periodic iff $\theta/2\pi \in \mathbb{Q}$ and is dense on the circle centered at $c$ otherwise.

3. *Neutral translation* ($\lambda = 1$):
   - $b = 0$: $x_t = x_0$ (every point is fixed).
   - $b \neq 0$: $x_t = x_0 + t\,b$ (unbounded linear drift).

4. *Expansive* ($|\lambda| > 1$): generically $|x_t| \to \infty$ exponentially; the unique nongeneric exception is $x_0 = c$ (then $x_t = c$).

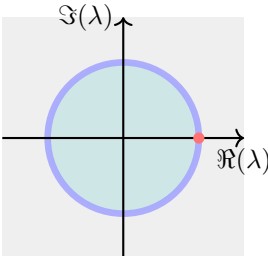

Figure 3: We distinguish 4 cases for $\lambda$: (teal) $\|\lambda\| < 1$, (purple) $\|\lambda\| = 1, \lambda \neq 1$, (pink) $\lambda = 1$, (gray) $\|\lambda\| > 1$.

**Boundedness.**  The trajectory $(x_t)$ is bounded if either $|\lambda| < 1$ or $|\lambda| = 1$ with ($\lambda \neq 1$) or ($\lambda = 1$ and $b = 0$). It is unbounded if $|\lambda| > 1$ (unless $x_0 = c$) or ($\lambda = 1$ and $b \neq 0$).

**Higher Dimensions.**  If $\Lambda \in \mathbb{C}^{d \times d}$ is diagonal and $b \in \mathbb{C}^d$, the dynamics decouple coordinate-wise and the above 1D classification applies to each coordinate independently.

# B  THE EXAMPLE OF THE GROUP $S_3$

## B.1  CAYLEY TABLE FOR $S_3$

| $\cdot$ | $e$ | $(12)$ | $(13)$ | $(23)$ | $(123)$ | $(132)$ |
|---|---|---|---|---|---|---|
| $e$ | $e$ | $(12)$ | $(13)$ | $(23)$ | $(123)$ | $(132)$ |
| $(12)$ | $(12)$ | $e$ | $(132)$ | $(123)$ | $(13)$ | $(23)$ |
| $(13)$ | $(13)$ | $(123)$ | $e$ | $(132)$ | $(23)$ | $(12)$ |
| $(23)$ | $(23)$ | $(132)$ | $(123)$ | $e$ | $(12)$ | $(13)$ |
| $(123)$ | $(123)$ | $(13)$ | $(23)$ | $(12)$ | $(132)$ | $e$ |
| $(132)$ | $(132)$ | $(23)$ | $(12)$ | $(13)$ | $e$ | $(123)$ |

Table 3: Cayley table of the symmetric group $S_3$.

## B.2  TWO-AUTOMATON CORRECTLY SIMULATES $S_3$

**Example 2.** Consider the $s^\alpha r^\beta$ encoding for $S_3$ elements, with $s = (12)$, $r = (123)$, $\alpha \in \{0,1\}$ and $\beta \in \{0,1,2\}$, and the transition rules described above. We represent the three states $Q_1, Q_2, Q_3$ of the second automaton by the cube roots of unity $e^{-i0} = 1$, $e^{-\frac{2i\pi}{3}}$, and $e^{-\frac{4i\pi}{3}}$. This representation does not change the fact that the automaton has discrete states; rather, it provides a convenient way to describe transitions as rotations, and later to connect the automaton to SSMs with continuous states. In this view, a rotation between states corresponds to multiplying by a discrete power of $e^{-\frac{2i\pi}{3}}$. Starting from the initial state $(1, e^{-i0})$, we apply each group element to obtain the automaton states and derive a decoding rule that maps any automaton state back to its associated group element.

For the identity $e$, the automaton remains $(1, 1)$. The swap $s = s^1 r^0$ flips $Q^{(1)}$ to $-1$ while leaving the second automaton unchanged, giving $(-1, 1)$, decoded as $s$. For $sr^2$, $Q^{(1)}$ flips to $-1$, and the second automaton rotates according to $Q^{(1)} = -1$ and $\beta = 2$, resulting in $(-1, e^{2i\frac{2\pi}{3}})$, decoded as $sr^2$. Similarly for $sr$, $Q^{(1)}$ flips to $-1$ and the second automaton rotates to $(-1, e^{i\frac{2\pi}{3}})$, decoded as $sr$. Finally, for $r$ and $r^2$, with no swap, the second layer rotates positively by $\frac{2\pi}{3}$ and $\frac{4\pi}{3}$, giving $(1, e^{-i\frac{2\pi}{3}})$ and $(1, e^{-i\frac{4\pi}{3}})$, decoded as $r$ and $r^2$.

Table 4 summarizes this decoding rule by providing a map between the states of the two-automaton and the elements of $S_3$.

| Automaton State | Group Element in $S_3$ |
|---|---|
| $(1, 1)$ | $e$ |
| $(-1, 1)$ | $s$ |
| $(-1, e^{i\frac{4\pi}{3}})$ | $sr^2$ |
| $(-1, e^{i\frac{2\pi}{3}})$ | $sr$ |
| $(1, e^{-i\frac{2\pi}{3}})$ | $r$ |
| $(1, e^{-i\frac{4\pi}{3}})$ | $r^2$ |

Table 4: Mapping between automaton states and group elements of $S_3$.

**Example 3.** Using the encoding, decoding, and transition rules above, one can in principle reproduce the full Cayley table of $S_3$ with the two-automaton. Here, we verify several nontrivial products to illustrate this behavior.

Trivial products such as $s \cdot s$ or $r \cdot r$ follow immediately. For less obvious cases, consider $sr^2 \cdot sr^2$. The first $sr^2$ maps the automaton to $(-1, e^{i\frac{4\pi}{3}})$; the second flips $Q^{(1)}$ back to 1 and

rotates the second automaton by $4\pi/3$ in the positive direction, returning the system to $(1,1)$, which is the correct result for two consecutive applications of the same swap. Similarly, $sr \cdot sr$ returns the automaton to $(1,1)$. Next, $s \cdot sr$ maps to $(-1,0)$ after $s$, then $sr$ flips $Q^{(1)}$ back to 1 and rotates the second automaton by $2\pi/3$, yielding $(1, e^{-i2\pi/3}) \equiv r$, with $\equiv$ denoting the equivalence. Reversing the order, $sr \cdot s$ gives $(-1, e^{i2\pi/3})$ after $sr$, then $s$ flips $Q^{(1)}$ back to 1, giving $(1, e^{i2\pi/3}) \equiv r^2$.

Finally, for $sr \cdot sr^2$, the first $sr$ gives $(-1, e^{i2\pi/3})$, and $sr^2$ flips $Q^{(1)}$ to 1 and rotates the second automaton by $4\pi/3$, yielding $(1, e^{-i2\pi/3}) \equiv r$. Conversely, $sr^2 \cdot sr$ maps first to $(-1, e^{i4\pi/3})$ and then to $(1, e^{i2\pi/3}) \equiv r^2$. These checks confirm that the two-automaton correctly reproduces the nontrivial entries of the Cayley table.

## C  PROOFS

### C.1  PROOF OF LEMMA 1

*Proof.* We split the proof into two cases:

(Case $|\lambda(x)_j| < 1$) In $M$, for all $g \in G$, we have $\lambda(\langle g \rangle \langle x \rangle^n)_j = \lambda(g)_j \lambda(x)_j^n \approx 0$ for sufficiently large $n$, where $\approx$ means equality at finite precision. We pick a sufficiently large $n$ such that $x^n = e$ as well. Note that this is always possible for all group elements $x$. For this $n$, the input sequence $\langle g \rangle \langle x \rangle^n$ transitions $h_j$ into $x$'s center of rotation (*i.e.,* fixed point) in coordinate $j$. Let $c$ be this center of rotation. We thus construct a new SSM $\widetilde{M}$ with $\tilde{\lambda}(g) = \tilde{\lambda}(gx^n) := \lambda(\langle g \rangle \langle x \rangle^n) \approx_j 0$ and $\tilde{b}(g) = \tilde{b}(gx^n) := b(\langle g \rangle \langle x \rangle^n) \approx_j c$ for all $g \in G$, where $\approx_j$ means that we are referring to the $j$th coordinate. In this new SSM, all inputs transition state-coordinate $j$ into the fixed point $c$ of input $x$. Note that $\widetilde{M}$ has been constructed in such a way that it respects the group law of $G$. Intuitively, this new SSM acts the way the old SSM would if we were to input $x$ for $n$ times after every $g$ seen in the input sequence and ignore the first $n$ outputs, and since $x^n = e$, the output of the new SSM should be the same.

(Case $|\lambda(x)_j| > 1$ or $|\lambda(x)_j| = 1 \wedge b(x)_j \neq 0$) With a similar argument to the previous case one can construct a new SSM where all inputs transition state-coordinate $j$ into `inf`, that is, the constant $c$ will be equal to `inf`. We are assuming here that in our finite-precision model, whenever the magnitude of a variable grows beyond some threshold, the variable gets fixed to a value, denoted by `inf`, that represents infinity. □

### C.2  PROOF OF LEMMA 2

*Proof.* With some simple algebra we get

$$\lambda^*\big(\lambda(h - c_1) + c_1 - c_2\big) + c_2 = (\lambda\lambda^*)h - (\lambda\lambda^*)c_1 + \lambda^*(c_1 - c_2) + c_2 \tag{12}$$
$$= h + (\lambda^* - 1)c_1 + (1 - \lambda^*)c_2 \qquad (\lambda\lambda^* = 1) \tag{13}$$
$$= h + (1 - \lambda^*)(c_2 - c_1). \tag{14}$$

Thus, the composition simplifies to $h \mapsto h + (1-\lambda^*)(c_2 - c_1)$, which is a non-zero translation since $\lambda^* \neq 1$ and $c_1 \neq c_2$. □

### C.3  PROOF OF LEMMA 3

*Proof.* We can find $\alpha_1, \alpha_2 \in \mathbb{N}$ such that $\lambda(g_1)_j^{\alpha_1} \lambda(g_2)_j^{\alpha_2} \approx 1$ at finite precision, $\lambda(g_1)_j^{\alpha_1} \neq 1$, and $\lambda(g_2)_j^{\alpha_2} \neq 1$. If the rotations are rational, this can be done exactly; if not, it can be done to arbitrary precision. Note that $\langle g_1 \rangle^{\alpha_1}$ and $\langle g_2 \rangle^{\alpha_2}$ induce neutral rotations about distinct centers in coordinate $j$. Thus, according to Lemma 2, the input sequence $\langle g_1 \rangle^{\alpha_1} \langle g_2 \rangle^{\alpha_2}$ induces a non-zero translation in coordinate $j$. Similar to Lemma 1, repeating this sequence causes the SSM to diverge to `inf`. □

### C.4  PROOF OF THEOREM 1

*Proof of Theorem 1.* ($\Rightarrow$) We assume there is a single-layer DCD SSM that tracks group $G$ and we show that $G$ is Abelian. Applying Lemma 1 to all coordinates implies that there exists a DCD

SSM layer $\widetilde{M}$ that tracks $G$ at finite precision, where no input and no coordinate has contraction or expansion or translation dynamics. In other words, all inputs and all coordinates have neutral rotation dynamics. Lemma 3 then implies that all inputs must induce neutral rotations *about the same center* in each coordinate. As a result, the effect of any input sequence is independent of the order of the inputs, and hence the group $G$ must be Abelian.

($\Leftarrow$) We assume $G$ is Abelian and we show there is a single-layer DCD SSM that tracks $G$. We construct a single-layer DCD SSM that tracks $G$. By the fundamental theorem of finite Abelian groups (Rotman, 2012), $G$ is isomorphic to a product of $n$ cyclic groups $C_{k_1} \times ... \times C_{k_n}$ for some $n$ and $k_1, ..., k_n$. Every group element $g$ can be represented as $(m_1, ..., m_n) \in [k_1] \times ... \times [k_n]$. The group's diagonal complex matrix representation $g \in G \mapsto \Lambda(g) \in \mathbb{C}^{n \times n}$, where $\Lambda(g)_{j,j} = \exp(2\pi i \frac{m_j}{k_j})$, can be used as the SSM transition matrix. With enough precision bits, $k_j$ roots of unity can be distinguished for all $j$. Let $h_0 = \mathbf{1}$ and $b(x) = \mathbf{0}$. The decoder simply maps each state to the corresponding group element. This SSM tracks $G$ at finite precision. $\qquad\square$

## C.5 Proof of Lemma 4

*Proof.* We do a proof by induction. The base case is the single-layer case ($k = 1$), which says that there exists another single-layer DCD SSM that also tracks group $G$ at finite precision, where, for all state-coordinates $j \in [d]$, the transition dynamics is fixed or a neutral rotation about a fixed center. We have already proved this in Theorem 1.

Let's assume the claim is true for $k \leq r - 1$ layers, and the goal will be to prove it for $k = r$. Let $\widetilde{M}^{(r-1)}$ be the SSM with the first $r - 1$ layers simplified. We apply the same strategy as the single-layer case in Lemma 1. Arbitrarily fix a coordinate $j \in [d]$ for the $r$th layer and the states of the first $r - 1$ layers of $\widetilde{M}^{(r-1)}$ to $h^{(1:r-1)}$. If some input sequence $\bar{x}$ keeps $h^{(1:r-1)}$ fixed and induces a contraction or expansion or translation in the $j$th component of the $r$th layer, with similar arguments as the single-layer case, we can construct another SSM where $\tilde{\lambda}^{(r)}(h^{(1:r-1)}, g)_j \approx 0$ for all $g \in G$. Otherwise, we skip this $j, h^{(1:r-1)}$ pair as it already satisfies the claim. Intuitively, this new SSM acts the way the old SSM would if every time the state of the first $r-1$ layers was $h^{(1:r-1)}$ we would input $\bar{x}$ for $n$ times, where $n$ is sufficiently large and such that the sequence $\bar{x}$ evaluates to identity, and ignore the first $n$ outputs. We repeat this procedure for all $j, h^{(1:r-1)}$ pairs, which are finite due to the finite precision assumption. Call the final SSM $\widetilde{M}^{(r)}$. $\qquad\square$

## C.6 Proof of Theorem 2

*Proof of Theorem 2.* ($\Rightarrow$) We assume there is a $k$-layer DCD SSM $M$ that tracks $G$ at finite precision and we show that $G$ admits a subnormal series of length $k$ with Abelian factors. First step of the proof is to replace the SSM with another SSM $M'$ that tracks the group according to Lemma 4. We now do a layer peeling argument starting from the first layer.

Due to finite precision, the state space is finite. There thus exists a finite input sequence that drives the system into a strongly connected component $\mathcal{C}$.[5] Let $g$ be the product of this sequence. For any $h \in G$, appending $g^{-1}h$ yields running product $h$ within $\mathcal{C}$, so $\mathcal{C}$ contains at least one state decoding to each group element. Since the model is exact at all input lengths, it correctly tracks $G$ within $\mathcal{C}$. We work within $\mathcal{C}$ from this point.

**Constructing $G_{k-1}$.** For a first-layer state $x_1$, define (overload) $\text{dec}(x_1)$ as the *set* of group elements decodable from some state in $\mathcal{C}$ with first-layer component $x_1$. We show these decoding sets are *either disjoint or equal*.

Suppose $g \in \text{dec}(x_1) \cap \text{dec}(y_1)$. Then there exist states $s, s'$ in $\mathcal{C}$ with first-layer components $x_1, y_1$ respectively, both decoding to $g$. By strong connectivity, there exists a path from $s$ to $s'$. Since both decode to $g$, this path has product $e$. So $\text{dec}(x_1) \subseteq \text{dec}(y_1)$. Similarly, there exists a path from $s'$ to $s$ with product $e$. So $\text{dec}(y_1) \subseteq \text{dec}(x_1)$.

The decoding sets thus partition $G$. Let $G_{k-1} = \text{dec}([x_e])$ where $[x_e]$ is the equivalence class of first-layer states containing the identity. We show $G_{k-1}$ is a subgroup. Take $g, h \in G_{k-1}$. Starting

---

[5]In fact, one can show that the state space is strongly connected to begin with but we don't need that.

from a state decoding to $h$ (first-layer in class $[x_e]$), append $h^{-1}g$. The running product is $g \in G_{k-1}$, so we land in class $[x_e]$. Hence $h^{-1}g$ maps class $[x_e]$ to itself. In particular, starting from $e \in G_{k-1}$ and appending $h^{-1}g$, the running product $h^{-1}g$ lands in class $[x_e]$, so $h^{-1}g \in G_{k-1}$. It is easy to show with similar arguments that $\mathrm{dec}([x_*])$ for any class is a coset of $G_{k-1}$. See Figure 4 for an example.

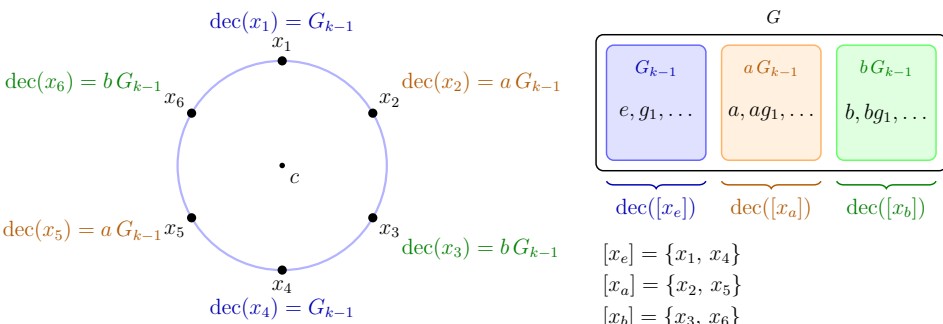

Figure 4: A simple example where the first-layer state is in $\mathbb{C}^1$, the group $G$ is partitioned into 3 cosets, and each equivalence class has size 2.

**Normality.** For $g_1 \in G_{k-1}$ and $a \in G$, the input sequences $(a, g_1, a^{-1})$ and $(g_1, a, a^{-1})$ produce the same accumulated rotation on the first layer (since rotations commute). The second has product $g_1 \in G_{k-1}$; the first has product $ag_1a^{-1}$. Since they land in the same first-layer class, $ag_1a^{-1} \in G_{k-1}$. Thus $G \trianglerighteq G_{k-1}$.

**Abelian quotient.** Similarly, $(a, b)$ and $(b, a)$ produce the same first-layer rotation for any $a, b \in G$, so $ab$ and $ba$ lie in the same class, *i.e.*, $abG_{k-1} = baG_{k-1}$.

**Peeling off.** Fix a first-layer state $x_0 \in [x_e]$. Let $\mathcal{X}$ be the set of input sequences that fix $x_0$. We claim that the set of products of these sequences is exactly $G_{k-1}$. The forward inclusion is immediate. For the reverse, take $g \in G_{k-1}$. Starting from a state $s_0$ with first-layer $x_0$ decoding to $e$, apply $g$ to reach state $s_1$ with first-layer $y_1$ (same class) decoding to $g$. Since $g \in \mathrm{dec}([x_0])$, there exists a state $s_2$ in $\mathcal{C}$ with first-layer $x_0$ decoding to $g$. By strong connectivity, there is a path from $s_1$ to $s_2$ with product $e$ (both decode to $g$). Concatenating gives a sequence with product $g$ taking $x_0 \rightarrow y_1 \rightarrow x_0$.

We now restrict to inputs from $\mathcal{X}$ and thus fix the first layer to $x_0$. Note that $\mathcal{X}$ is closed under concatenation. With the first layer fixed, Lemma 4 gives layer 2 a fixed center and inputs (from $\mathcal{X}$) cause rotations on layer 2. We can now apply the same argument as we did for the first layer to obtain $G_{k-1} \trianglerighteq G_{k-2}$ with $G_{k-1}/G_{k-2}$ Abelian. Iterating over all $k$ layers produces the chain.

**Termination.** After $k$ layers, $G_0$ consists of elements whose input sequences fix all layers. Since the model is exact, distinct elements produce distinct states, so $G_0 = \{e\}$.

($\Longleftarrow$) We assume there is a subnormal series $(G = G_k) \trianglerighteq G_{k-1} \trianglerighteq \ldots \trianglerighteq G_1 \trianglerighteq (G_0 = \{e\})$ where $G_{i+1}/G_i$ is Abelian for all $i \in [k]$ and we show that there exists a $k$-layer DCD SSM that tracks $G$. We do a proof by induction on $k$. The base case ($k = 1$) is the single-layer setting which we have already proved in Theorem 1.

We now assume that we have a $k - 1$ layer DCD SSM that tracks the group $N := G_{k-1}$ which is a normal subgroup of $G$. We show that we can add an initial layer to get a $k$-layer DCD SSM that tracks $G$. The first layer is constructed following Theorem 1 to track the Abelian group $H := G_k/G_{k-1}$.

We now describe how the first layer interacts with the top $k - 1$ layers. Fix a section $s : H \rightarrow G$ of the quotient map $G \rightarrow H$. Every $g \in G$ can be written as $g = n\,s(h)$ for some $n \in N$ and $h \in H$, and we use this representation for the SSM state. For an input token we write $g = s(h)\,n$ and assume it right-multiplies the state.

Let the current state be $g' = n'\,s(h')$ and the input be $g = s(h)\,n$. Then

$$g'g = n'\,s(h')\,s(h)\,n. \tag{15}$$

Since $s(h')s(h)$ and $s(h'h)$ lie in the same coset of $N$, there exists $d(h', h) \in N$ such that

$$s(h')\, s(h) = d(h', h)\, s(h'h). \tag{16}$$

Substituting and inserting $s(h'h)^{-1}s(h'h)$ on the right gives

$$g'g = n'\, d(h', h)\, \left(s(h'h)\, n\, s(h'h)^{-1}\right) s(h'h). \tag{17}$$

Because $N \trianglelefteq G$, the conjugated term $s(h'h)\, n\, s(h'h)^{-1}$ lies in $N$. Thus the updated state has the form $n'\kappa(h', g)s(h'h)$, where $\kappa(h', g) = d(h', h)(s(h'h)ns(h'h)^{-1}) \in N$. This means that the first layer should update its own state by applying $h \in H$ and output $\kappa(h', g) \in N$ as input to the top $k - 1$ layers to achieve the overall desired update of group $G$. See Figure 5.

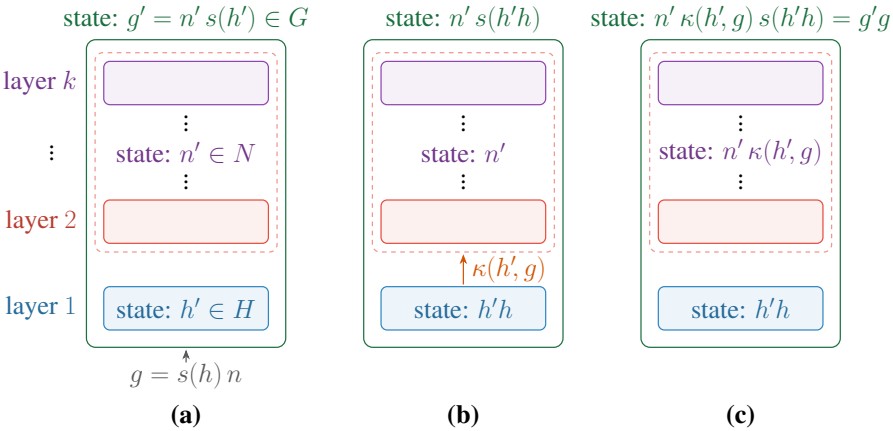

Figure 5: **(a)** Model is in state $g'$ and receives input $g$. **(b)** First layer updates from $h'$ to $h'h$ and outputs $\kappa(h', g)$. **(c)** Higher layers update from $n'$ to $n'\kappa(h', g)$. The final collective state is $n'\kappa(h', g)s(h'h) = g'g$ as desired.

$\square$

