# OpenReview forum: "The Expressive Limits of Diagonal SSMs for State-Tracking"
_ICLR.cc/2026/Conference — ICLR 2026 Poster_

### Official Review · Reviewer_MdGT · 2025-10-31

**Soundness:** 4
**Presentation:** 4
**Contribution:** 4
**Rating:** 8
**Confidence:** 3

**Summary:**

This paper provides a theoretical characterization of what diagonal State-Space Models (SSMs) can and cannot learn when tracking sequential state information. The main results can be summarized as
    1. A single diagonal SSM layer with complex values can track any Abelian group but cannot track any non-Abelian group.
    2. A k-layer diagonal SSM can track a group if and only if that group has a "subnormal series" of Abelian factor groups with length ≤ k. This means stacking layers expands expressivity to solvable groups in a precise, depth-dependent way.
    3. While multi-layer models are theoretically expressive enough for certain non-Abelian groups, they consistently fail to learn these solutions in practice with standard gradient descent.

**Strengths:**

The notations and logical flow is very clear make it easy to follow the results.
This work provides a complete characterization of diagonal SSM expressivity and exposes a critical gap between representational capacity and optimization.
Diagonal SSMs form a strict subset of the computational hierarchy: sufficient for Abelian group structures but provably insufficient for non-Abelian groups at single-layer depth, with depth providing only limited help in practice.

**Weaknesses:**

The results are pure theoretical, it would be good if there is any connections to real applications.

**Questions:**

The theoretical contribution is significant. I'm interested in whether there are any real applications that can make use of this results. For example in material science, there may exists certain task that the symmetry property may exists?

---

> ### Author Response · Authors · 2025-11-21
>
> We thank the reviewer for their review and positive assessment of our work. Below we respond to the question and list some possible real applications of state-tracking:
> 1. Material science: The parity of electron counts is very important and is related to properties such as chemical reactivity and magnetic properties. Also, in conjugated ring systems, mod-4 parity of electron counts is related to aromaticity. Therefore, better state-tracking performance might lead to better performance on material science tasks that rely on the aforementioned properties. This area is outside of our expertise but seems like a very exciting direction for future work.
> 2. Next-frame prediction for world modeling of a video of someone permuting identical cups under which different items have been hidden. In order to correctly predict the next frame when the items are revealed, the model must have been able to perform state tracking of the group $S_3$ internally.
> 3. Next-token prediction for a chess game written down in standard chess notation. The chessboard is a latent state that is being manipulated by the chess moves. In order to know which moves are legal the model must be able to internally keep track of the state of the chess board.
> 4. The task of answering questions about behavior of a given Python code. See for example our response to weakness 3 of reviewer 644N.

---

> > ### Comment · Reviewer_MdGT · 2025-11-26
> >
> > I appreciate the author for the reply. Thanks for addressing my main concerns, I will maintain my score.

---

### Official Review · Reviewer_644N · 2025-10-31

**Soundness:** 3
**Presentation:** 3
**Contribution:** 2
**Rating:** 4
**Confidence:** 2

**Summary:**

This paper studies the theoretical expressivity of diagonal State-Space Models (SSMs) on group state-tracking tasks. They show that a single-layer input-dependent complex-valued diagonal (DCD) SSM can track a group G at finite precision if and only if G is Abelian. A $k$-layer DCD SSM can track a group G if and only if G has a subnormal series of length at most $k$ with Abelian factor groups. The authors show a gap between this theory and their experiments. While the theory shows multi-layer diagonal SSMs can express solvable non-Abelian groups, experiments demonstrate they struggle to learn these solutions in practice. Even 2-layer models that can theoretically represent S3 fail to learn generalizable solutions.

**Strengths:**

1. The paper provides necessary and sufficient conditions for when diagonal SSMs can track groups, not just sufficient conditions or impossibility results
2. The connection to group theory seems an elegant way to relate architectural constraints to algebraic properties.
3. The paper doesn't just prove existence - it shows explicit constructions demonstrating how multi-layer diagonal SSMs can track non-Abelian groups.
4. The experimental section reveals an important gap between expressivity and learnability
5. The paper carefully handles finite precision constraints, which are practically relevant and often glossed over in theoretical work.
6. The results apply to popular SSM variants like Mamba

**Weaknesses:**

1. The experiments only test on 5 groups and don't explore what makes some solvable groups learnable vs others. More extensive experiments would strengthen claims about the learnability gap.
2. While the paper identifies that multi-layer models fail to learn non-Abelian groups, it doesn't deeply investigate why or propose solutions beyond noting "optimization difficulties"
3. State-tracking is a specific synthetic task family. The paper doesn't clearly connect these limitations to practical sequence modeling tasks
4. The paper doesn't compare with non-diagonal SSMs empirically, which would help quantify the cost of the diagonal constraint in practice.
5. While mentioning block-diagonal structures could help, the paper doesn't explore intermediate architectures between fully diagonal and fully dense.
6. The paper could better position results relative to circuit complexity findings ($TC^0$) and explain what new understanding this group-theoretic view provides--that was not very clear to me.

**Questions:**

1. Which real-world sequence modeling tasks actually require tracking non-Abelian groups? I'm not familiar enough with the matter to know how common these requirements are in NLP and other related domains
2. Can you identify any specific optimization challenges when learning non-abelian groups? (e.g. loss landscape, initiatlization?) Given the explicit construction for S3, could we initialize models closer to theoretical solutions to improve learnability?
3. What's the minimal architectural change needed to make non-Abelian groups learnable? Would 2×2 block-diagonal suffice for S3?
4. How do these limitations apply to transformers, if at all?

---

> ### Author Response · Authors · 2025-11-21
>
> We thank the reviewer for their extensive review and the many mentioned strengths. Below we address weaknesses and questions.
>
> *Weaknesses:*
>
> 1. Thanks for the suggestion. We initially selected 5 groups to report which cover the different kinds of groups that we are interested in: parity ($C_2$), a small and a large Abelian group ($C_6$ and $C_{60}$), two small solvable non-Abelian groups with subnormal chains of length 2 ($S_3$ and $A_4$). We tried other groups as well in our own experiments, but did not find anything new or different that would be worth reporting. For the rebuttal, we have extended our table and added 4 more groups: the cyclic group $C_{24}$, two products of cyclic groups $C_2 \times C_4$ and $C_3 \times C_6$, and the non-solvable group $A_5$. We observe that DCD SSMs are able to learn and extrapolate on these newly introduced Abelian groups as expected. We also see that no model can learn $A_5$.
>
> 2. Please see response to question 2.
>
> 3. An example of a sequence modeling task that requires keeping track of a state that is being manipulated is program state analysis from big-bench (https://github.com/google/BIG-bench/tree/main/bigbench/benchmark_tasks/auto_debugging). For example, given the following code
> ```
> x, y, z = 0, 1, 2
> x, y = y, x
> y, z = z, y
> x, y = y, x
> ```
> What is the value of `x` after line 4?
> In this specific example it is important to be able to model the group $S_3$ of permutations of 3 objects. We will add this example to the paper and elaborate more on the connections to practical sequence modeling tasks.
>
> 4. We have experimented with IDS4 [Merrill et al., 2024] which is an SSM with a full transition matrix. However there are instability problems with IDS4 which prevents the model from learning effectively and generalizing. Due to this difficulty, IDS4 does not show any extrapolation on any of the studied groups, even $C_2$. Hence, we did not include it. To the best of our knowledge, there are no other SSMs with input-dependent full transition matrices. Nevertheless, we note that RNN is an example of a recurrent model with full transition matrix (and non-linear recurrence). While non-linearity has a big effect on expressivity, maybe partly RNN’s strong performance is also due to its full transition matrix.
>
> 5. Studying the deviation from diagonality was out of the scope of our work. This is a very interesting point that has been studied in [Grazzi et al., 2024]. They quantify the trade-off between deviation from diagonality and the complexity of the group to track, by parameterizing the transition matrix as a diagonal plus rank-n matrix. Here, we are interested in keeping the transition matrix diagonal and instead find the minimum number of layers of the diagonal SSM for solving a subset of the group state tracking tasks, i.e., the solvable ones. Nevertheless, it will be an interesting future direction to compare the efficacy and efficiency of these two approaches.
>
> 6. The circuit complexity approach only says that non-solvable groups are beyond the expressivity of diagonal SSMs, conditional on $\mathsf{TC}^{0} \neq \mathsf{NC}^{1}$ (which is widely conjectured to be true). On the other hand, our results provide fine-grained expressivity of k-layer diagonal SSMs and also do not depend on any conjecture. Before our work and using only the circuit complexity approach, it was not known that single-layer diagonal SSM cannot track non-Abelian groups at finite precision. Also, importantly, for a given solvable group, we specify a lower bound on the number of DCD SSM layers necessary for expressing its state-tracking solution.

---

> ### Author Response · Authors · 2025-11-21
>
> *Questions:*
> 1. Consider tracking the position of an agent on a 2D grid that wraps around. The agent can move forward/backward or turn left/right. Turning left and then moving forward is not the same as moving forward and then turning left, so the underlying group is not Abelian. Suppose the agent has to move to a goal and the model needs to tell us if the agent has arrived at the goal or not. The inputs to the model are a list of the agent’s actions. One can think of this as an abstract and simplified world-modeling task. Similar capabilities are likely needed in more real-world settings where the underlying groups are hidden beneath a lot of details and complexity.
> 2. Thanks for the suggestion. Initializing AUSSM close to the solution does help and the model successfully learns and extrapolates to at least four times longer sequence lengths. We will mention this in the paper. This gives us some insight into the loss landscape and suggests that solutions lie within a basin of attraction. However, the rest of the loss landscape is still not well understood. A work of the nature of Hahn & Rofin (2024) (which is for transformers), but for SSMs, would be a very nice future work direction for our paper. However, since initializing close to the solution is helpful, the optimization challenges of SSMs might be very different from those of transformers, where solutions are isolated points in weight-space. We believe further research is needed in this area.
> 3. Yes, complex 2x2 blocks would allow $S_3$ to be learned by a single layer. It would also put some non-solvable groups, e.g., the simple non-Abelian group $A_5$, within the model’s expressive range. A paper on circuit complexity worth mentioning is Mereghetti & Palano (2000) which uses the group LF(2, 5), which is isomorphic to $A_5$, to show that iterated matrix multiplication of 2x2 matrices is in $\mathsf{NC}^{1}$.
> 4. By the state-space duality of Gu & Dao (2024) perhaps one could show that the same limitations apply to sliding window transformers with some kind of learned input-dependent ROtational Positional Embeddings (ROPE) but we have not looked at this closely.
>
> *References:*
>
> [Mereghetti & Palano, 2000] Threshold circuits for iterated matrix product and powering (RAIRO: ITA, 2000)
>
> [William Merrill, Jackson Petty, Ashish Sabharwal] The Illusion of State in State-Space Models (ICML, 2024)
>
> [Hahn & Rofin, 2024] Why are Sensitive Functions Hard for Transformers? (ACL, 2024)
>
> [Grazzi et al., 2024] Unlocking State-Tracking in Linear RNNs Through Negative Eigenvalues (ICLR 2025)
>
> [Gu & Dao, 2024] Transformers are SSMs: Generalized Models and Efficient Algorithms Through Structured State Space Duality (ICML 2024)

---

### Official Review · Reviewer_1jQf · 2025-10-31

**Soundness:** 3
**Presentation:** 3
**Contribution:** 2
**Rating:** 4
**Confidence:** 2

**Summary:**

The paper studies the expressivity of input dependent, complex valued diagonal state space models. Is shows that under some mild decoder conditions, the single layer can track any abelian group but not any non-abelian group. The paper performs empirical experiments that show that while multi-layer DCD SSMs are theoretically expressive enough for non-abelian groups, they fail to learn it in practice.

**Strengths:**

The paper improves the understanding of the expressivity of State Space Models (SSM), by focusing on a specific type of SSM and a particular data type (abelian vs non-abelian groups). This seems to be a fresh perspective on analyzing expressivity and the theory appears to be rigorous.

**Weaknesses:**

One concern is on the significance of the result. Does this make the SSM architecture more expressive (or less) than a transformer? Does this have any practical implications on how we should train SSMs?

Another point that makes it harder to understand the significance of the theory is that the experimental results do not directly support the theory but instead suggest that the finding that even if the models theoretically can learn certain tasks, the optimization fails to do it. This is not in itself bad, but the paper would have been much stronger if there were empirical results that supported the theory.

For the task C_60, for instance, it's not clear how the training task is generated. Is the input 1,2,3,4,..., or is it a random draw of numbers between 1 and 59, or something else.

Minor:
Somewhat unusual formatting with the theorem boxes.
I suspect that many people who are experts on SSMs may not be deeply familiar with group theory. Hence giving concrete example to show what for instance, C_60, is may allow a broader audience enjoy the paper.

**Questions:**

Is there some relevant real world task that correspond to non-abelian group?

---

> ### Author Response · Authors · 2025-11-21
>
> We thank the reviewer for their constructive review. Below we address weaknesses and questions.
>
> *Weaknesses:*
> * Expressivity of Transformers: DCD SSMs are more expressive than transformers at finite precision. Hahn (2020) shows that, for the task of parity, the output distribution of a transformer at position n goes to uniform as n grows. At finite precision, this means that transformers cannot solve parity for arbitrary lengths as for large enough n the output will be 50/50, i.e., random guess. At infinite precision, on the other hand, Chiang & Cholak (2022) show that transformers *can* solve parity, but they observe that these solutions are hard to learn. To explain the difficulty of learning parity, Hahn & Rofin (2024) study the loss landscape of transformers and show that high-sensitivity solutions are isolated points in weight-space that are surrounded by low-sensitivity solutions, leading transformers to favor low-sensitivity solutions. Note that parity is an example of a high-sensitivity function, since changing the first input token affects the output at arbitrarily far positions in the future. A work of the nature of Hahn & Rofin (2024), but for SSMs, would be a very nice future work direction for our paper.
> * Implications of our work: As for practical implications for training SSMs, our results do not imply any specific training instructions, but instead they specify minimum requirements from the diagonal SSM architecture, such as the number of layers. More specifically, independently of the learnability, we conclude that for a diagonal SSM to solve a task that requires tracking a non-Abelian group with at least $k$ Abelian factors in its subnormal chain, the SSM model must have at least $k$ layers.
> Overall, the significance of our results is that we made a first step by rigorously categorizing the limits of diagonal SSMs on tasks requiring state tracking capabilities. This then guides us on where to focus efforts to explore learnability issues, i.e., not on the cases where the origin of the problem is known to be the expressivity. Training SSMs to reach their theoretical expressivity is an issue that we highlight in this paper and encourage future work on.
> Our work also identifies the source of expressivity or lack thereof in SSMs and provides insights that can lead to architectures that track a broader family of groups.
> * Supporting the theory: As we show, it is possible to set the weights of an SSM such that it solves the state-tracking tasks that are theoretically possible, for example, $S_3$ for a 2-layer SSM, where learning fails with gradient descent. So a solution exists in the weight space, but it is not discovered with gradient descent. While one cannot “verify” our impossibility results empirically, our results provide one data point in support of the theory. The most important support for our claims is, of course, the proofs.
> We’ve also found that initializing close to the solution (per the suggestion of reviewer 644N) is helpful and successfully allows length-extrapolation for $S_3$ with AUSSM. This suggests that further research might produce a training or architectural novelty which allows learning non-Abelian groups with DCD SSMs.
> * Task $C_{60}$: The inputs are chosen uniformly at random from the group. We will clarify this in the paper. The task is essentially addition mod 60. An example input for this task is the sequence [51, 20, 4, 49] and the correct output is [51, 11, 15, 4]. Thanks for the suggestion. We will add this example to the paper to improve readability.
>
> *Questions:*
> Please see the response to Q1 of reviewer 644N.
>
> *References:*
>
> [Hahn, 2020] Theoretical Limitations of Self-Attention in Neural Sequence Models (TACL, 2020)
>
> [Chiang & Cholak, 2022] Overcoming a Theoretical Limitation of Self-Attention (ACL, 2022)
>
> [Hahn & Rofin, 2024] Why are Sensitive Functions Hard for Transformers? (ACL, 2024)

---

### Official Review · Reviewer_6RjB · 2025-10-31

**Soundness:** 2
**Presentation:** 3
**Contribution:** 3
**Rating:** 6
**Confidence:** 3

**Summary:**

This paper demonstrates the limitations of input-dependent complex diagonal SSMs in terms of expressiveness for groups state tracking. The authors show that a single layer can track Abelian groups, and k layers can solve up to depth k; thus a two-layer model can track S_3. The paper also shows that diagonal models are expressive enough in theory, but in fact hard to train on non-Abelian tasks.

**Strengths:**

The authors provide clear iff theorems and link depth with group structure: i.e., 1-layer - Abelian, and k-layer - length <= k. The paper also provides a good demonstration of two-layer S_3 that illustrates the theory, and presents an interesting observation that expressivity does not directly lead to learnability in reality.

**Weaknesses:**

- It is uncertain if the observations in this paper will directly lead to same results in real-world benchmarks such as language modeling.
- the experiments in the paper are limited, not providing results with different state dimensions, precisions, and decoders.
- it is uncertain how the training details in the paper, and unclear if it is actually true that expressivity != learnability.

**Questions:**

- How many, and what different training settings have the authors tried?
- Do the results look the same no matter how the settings (e.g., hyperparam, learning rate, weight decay, scheduling) change?
- how crucial is the universal decoder?

---

> ### Author Response · Authors · 2025-11-21
>
> We thank the reviewer for their thoughtful assessment and for the detailed questions and suggestions. Below we respond to the questions and weeknesses.
>
> Weaknesses:
>
> * It is true that higher expressivity does not necessarily lead to better performance on downstream tasks. However, previous and concurrent works (Grazzi et al. 2024, Siems et al., 2025, Mamba-3) show that modifying the architecture to improve state-tracking expressivity sometimes leads to improvements on various language modeling and downstream tasks. See, for example, Fig. 5 of [Grazzi et al., 2024] and Fig. 1 and Fig. 8 of [Siems et al., 2025], which show improvements on coding (CodeParrot benchmark) and math (MathHard and OpenThoughts 114K Math benchmarks). We believe state-tracking expressivity brings valuable insight for architecture design while also acknowledging that performance on downstream tasks depends on many intertwined factors that are not yet fully understood.
>
> * We ran the experiments with standard FP32 precision. We had also tried various hyperparameters in our experiments with the AUSSM model on $S_3$ and $A_4$, including different learning rates, state dimensions, and architectures for the decoder. Regardless of the setting, AUSSM always fails to extrapolate for these groups. For the rebuttal, we added extensive experiments for all models on all groups, with different state dimensions (32, 64, 128), learning rates (.0001, .0005, .001), and learning rate schedulers (no scheduler, cosine and reduce-on-plateau). The new experiments improved the extrapolation of the models on the Abelian tasks (we updated Table 2 based on the new results), however, they did not change the significant observation that $S_3$ and $A_4$ could not be tracked by DCD SSMs. Hence, the final takeaway remains the same. We will edit the paper to add a list of settings that we tried and report the best length-generalization performance. We thank the reviewer for their constructive feedback.
>
> * The details of the experiments are in the experiment section of the paper. Please let us know if there is anything you would like us to add.  Regarding learnability vs. expressivity, we are confident in our results because we tested many model variants; however, we don’t rule out the possibility that architectural or training innovations might solve the issue in the future, and we hope to motivate such research with this work.
>
>
>
> Questions:
>
> * We explored various modifications to the architecture, learning rate, training curriculum, weight decay, and skip connections, among others, but did not observe successful length-generalization on $S_3$ or $A_4$ with any model. We ran our final experiments with the best settings found during our explorations. Additionally, we have performed a more systematic sweep for the rebuttal which will be added to the paper.
>
> * Yes. However, very bad hyperparameters can even cause the SSMs to fail to learn the Abelian groups.
>
>
> * The universal decoder is designed to strengthen the theoretical impossibility results. When we show that an SSM is not able to solve a certain state-tracking task, the issue is not the limited expressivity of the decoder, because we have assumed it to be universal; The issue is the limited expressivity of the SSM layers.
>
>
> *References:*
>
> [Grazzi et al., 2024] Unlocking State-Tracking in Linear RNNs Through Negative Eigenvalues (ICLR, 2025.)
>
> [Siems et al., 2025] DeltaProduct: Improving State-Tracking in Linear RNNs via Householder Products (NeurIPS 2025)
>
> [Mamba-3] Mamba-3: Improved Sequence Modeling Using State Space Principles (under review at ICLR, 2026)

---

> > ### Comment · Reviewer_6RjB · 2025-11-28
> > **Reply to the authors**
> >
> > I appreciate the authors for the answers to my questions and concerns. Just one request (suggestion) is, could the authors show the perplexity of these difference variants on a very small-sized language dataset (for instance, wikitext)? I totally understand that the rebuttal period is short, but I also believe showing the actual perplexity of different variants is the easiest and the most direct way to prove the claim of better expressivity.

---

### Author Response · Authors · 2025-11-21

General Response:

Dear reviewers, thank you for your extensive reviews and constructive feedback. Please find below the updated tables of results. We are now reporting best length-extrapolation over 3 seeds, state dimension in {32, 64, 128}, learning rate in {1e-4, 5e-4, 1e-3}, and learning-rate scheduler in {fixed, reduce learning-rate on plateau, cosine}. We have also added the Abelian groups $C_{24}$, $C_2 \times C_4$ and $C_3 \times C_6$, and the non-solvable group $A_5$. The main takeaway remains the same.


*Results for single-layer models:*
| Task               | Mamba  | Negative Mamba | Simple AUSSM  | AUSSM  | RNN  |
|--------------------|--------|----------------|---------------|--------|------|
| $C_2$              | ✗      | 1000           | 160           | 1000   | 1000 |
| $C_6$              | ✗      | ✗              | 240           | 940    | 1000 |
| $C_{24}$           | ✗      | ✗              | 240           | 260    | 1000 |
| $C_{60}$           | ✗      | ✗              | 300           | 240    | ✗    |
| $C_2 \times C_4$   | ✗      | ✗              | 140           | 200    | 1000 |
| $C_3 \times C_6$   | ✗      | ✗              | 500           | 200    | ✗    |
| $S_3$              | ✗      | ✗              | ✗             | ✗      | 1000 |
| $A_4$              | ✗      | ✗              | ✗             | ✗      | 1000 |
| $A_5$              | ✗      | ✗              | ✗             | ✗      | ✗    |


*Results for 2-layer models:*

| Task               | Mamba  | Negative Mamba | Simple AUSSM  | AUSSM  | RNN  |
|--------------------|--------|----------------|---------------|--------|------|
| $C_2$              | ✗      | 1000           | 1000          | 200    | 1000 |
| $C_6$              | ✗      | ✗              | 240           | 100    | 1000 |
| $C_{24}$           | ✗      | ✗              | 300           | 160    | 1000 |
| $C_{60}$           | ✗      | ✗              | 260           | ✗      | ✗    |
| $C_2 \times C_4$   | ✗      | 360            | 160           | ✗      | 1000 |
| $C_3 \times C_6$   | ✗      | ✗              | 260           | 200    | ✗    |
| $S_3$              | ✗      | ✗              | ✗             | ✗      | 1000 |
| $A_4$              | ✗      | ✗              | ✗             | ✗      | 1000 |
| $A_5$              | ✗      | ✗              | ✗             | ✗      | ✗    |


*Remark:* The group $C_4$ can be written as the subnormal chain $C_4 \triangleright C_2 \triangleright \set{e}$, where the factors are all $C_2$. This means that, according to our theory, stacking two layers, each of which is capable of solving $C_2$, e.g., Negative Mamba, can do state-tracking for $C_4$. And we see in practice that 2-layer Negative Mamba is able to do state-tracking for $C_2 \times C_4$. Interestingly, this is a case where the increased expressivity from stacking layers *is* usable through gradient-based learning. We will add these results and observations to the paper.

---

### Author Response · Authors · 2025-12-04

We would like to thank the reviewers once more for their insightful review and helpful feedback and questions.
Here are the changes made to our submission based on all reviews:

1- We updated the paper with what we answered the reviewers and promised to add.
More specifically, we added more experiments on a wider range of solvable groups and reported experimental results based on an extensive hyperparameter search. We explained in the paper that while gradient descent could not find a generalizable solution for the non-Abelian solvable group of $S_3$, initializing the training close to the analytical solution was effective and resulted in a *learned*  solution that could also extrapolate. We added more examples of real-world benchmarks where state tracking abilities are assumed to be essential to solve the task. Importantly, we better elaborated on positioning of our theoretical results relative to circuit complexity findings; more specifically, we emphasised on our significant novel result on the expressivity of multi-layer DCD SSM, which is a lower bound on its depth so that it can track non-Abelian solvable groups with a subnormal chain of a given length.

2- We simplified the theory by merging the first two lemmas, also we improved the clarity and flow of the proof for the second theory.

3- We clarified the choice of our groups and included some examples of groups to provide more context on the group theory content of the paper.

4- We restructured the Experiments section to better explain our experimental setup.


We hope that these changes address most of the reviewers' concerns and questions.

Regarding the important suggestion of Reviewer 6RjB, we have been running experiments on language modeling. On the Wikitext dataset, we did not observe any improvement by replacing (partially or for all layers) Mamba layers with AUSSM layers (until now, our observation is limited to the training perplexity, and we are still running experiments). However, we expect state tracking to be important in language modeling on benchmarks with code or math, such as CodeParrot. Due to the size of the dataset, we have not yet finished experiments on CodeParrot. We are planning to include both experiments in the final version of the paper, and we thank the reviewer for suggesting this experiment.

---

### Meta-Review · Area_Chair_SmRc · 2026-01-06

**Summary:**

This paper sheds light on the expressivity of input-dependent complex-valued diagonal state space models for sequential state-tracking tasks. The authors provide a good combination of theoretical and empirical results that advances our understanding of what diagonal state-space models can and cannot learn in this context.

All reviews are positive and thoughtful. The authors used the rebuttal phase effectively to address the reviewers’ questions and main concerns. In particular, they provided additional experiments, streamlined the presentation of the theoretical results, and improved the clarity and organization of the paper. The revised manuscript reflects a careful revision and addresses the key issues raised during review.

In summary, I find this to be a strong and interesting contribution that will likely motivate future work in the area. Given the positive reviews and the improvements made during rebuttal, I recommend accept.

**Reviewer Concerns:**

All main concerns have been addressed by the rebuttal.

**Reviewer Scores:**

The reviews were positive overall, and the rebuttal and discussion phase may have convinced the lower-confidence reviewers to slightly increase their scores.

---

### Decision · Program_Chairs · 2026-01-26

Accept (Poster)